# Comprehensive Geriatric Assessment (CGA) and Optimisation Services in Older Kidney Patients: Results from the First UK-Wide Transplant Centre and Renal Unit Survey Study

**DOI:** 10.3390/jcm14093070

**Published:** 2025-04-29

**Authors:** John A. Holland, Judith S. L. Partridge, Antonia J. Cronin

**Affiliations:** 1Department of Transplant Surgery, Renal Medicine, and Urology, Guy’s and St Thomas’ NHS Foundation Trust, London SE1 7EH, UK; 2School of Immunology and Microbial Sciences, Faculty of Life Sciences and Medicine, King’s College London, London WC2R 2LS, UK; 3Department of Perioperative Medicine for Older People undergoing Surgery, Guy’s and St Thomas’ NHS Foundation Trust, London SE1 7EH, UK; 4School of Population Health & Environmental Sciences, Faculty of Life Sciences and Medicine, King’s College London, London WC2R 2LS, UK

**Keywords:** frailty, multimorbidity, cognitive impairment, transplantation, equitable access, health inequalities, comprehensive geriatric assessment

## Abstract

**Background:** Demand for renal replacement therapy (including dialysis, transplantation and supportive care) in patients over 60 is increasing. Concerns regarding poorer outcomes and decision-making in this cohort have been raised. Evidence suggests these relate to frailty, multimorbidity and cognitive impairment, all seen frequently in older age. Comprehensive Geriatric Assessment (CGA) is a multidisciplinary methodology proven to improve outcomes relating to this triad and could be transformative for older kidney patients. This national UK survey aims to describe (1) attitudes/beliefs of renal physicians and transplant surgeons in the UK toward the CGA for older potential kidney transplant recipients and those being considered for dialysis or supportive care; (2) provision of CGA services for these patients in the UK; (3) barriers and enablers to the provision of these CGA services in the UK. **Methods:** The UK’s 72 renal units (RUs) and 23 adult kidney transplant centres (TCs) were invited to complete online surveys electronically using a protected link (24 April 2024–31 August 2024). **Results:** The response rate was 100%. Only six RUs offered CGA services. However, respondents overwhelmingly advocated CGA for older patients being considered for transplant (RUs 47/55, TCs 17/19), dialysis (RUs 52/54) and supportive care (RUs 51/54). Lack of funding to support CGA-OS (45/51), lack of available staff to deliver CGA (44/51) and time constraints (36/51) were reported barriers to implementing CGA by RUs. TCs identified lack of funding (13/18) and published evidence (12/18) as the main barriers. **Conclusions**: Transplant surgeons and renal physicians alike support CGA for older kidney patients, but only six UK units currently offer the service to these patients. Research developing and implementing CGA for this population is essential to optimise outcomes and influence policy at the national level.

## 1. Introduction

The global population is ageing. A recent United Nations report projects a doubling of those aged 65 years or older by 2050 [1]. Chronic kidney disease (CKD) is more prevalent in older people. Demand for renal replacement therapy (dialysis and transplantation), as well as supportive care options, will inevitably increase as the population ages.

Frailty; multimorbidity (including hypertension, diabetes and cardiovascular disease) and cognitive impairment are all observed more frequently with increasing age. This triad has been associated with poorer outcomes in patients undergoing dialysis and following transplant. Currently, they are regarded as relative barriers to transplantation, even though kidney transplantation is the treatment of choice for end-stage kidney disease (ESKD), both in terms of improved quality of life and cost-effectiveness, compared to dialysis [2,3].

Older adults with CKD are 10 times more likely to be frail than age-matched general populations [4]. Estimates suggest that 14% of non-dialysis CKD patients over 60 years and 71% of dialysis-dependent CKD patients over 65 years are frail [5]. Frail CKD patients over 60 are more likely to die and only half as likely to be transplanted within two years compared to their age-matched non-frail peers [6,7]. Moreover, when transplanted, frail patients are more likely to experience delayed graft function, longer hospital stays and all-cause mortality at 12 months post-transplant [8].

Multimorbidity, the coexistence of two or more long-term health conditions, is also prevalent in older patients with CKD. One multicentre UK study found that 96% of CKD patients (mean age 66) were multimorbid [9]. Multimorbidity is associated with increased peri-operative risk at the time of transplant and a higher mortality rate 12 months after transplant when compared to age-matched non-multimorbid CKD patients [10].

Cognitive impairment (CI) is frequently encountered in CKD patients aged over 60 years. Studies estimate a prevalence of up to 75% in CKD patients compared to only 20% in the age-matched general population [11]. CI in older patients can impair their decision-making capacity and affect transplant outcomes [12]. In the immediate post-transplant period, patients with CI are more likely to experience delirium, which is associated with increased length of hospital stay, risk of institutional discharge, graft loss and mortality at one year [13]. Longer-term CI is strongly associated with immunosuppression non-adherence and subsequent graft loss. in addition to the potential worsening of cognitive trajectory caused by postoperative delirium [14].

Currently, despite the many ways in which frailty, multimorbidity and cognitive impairment negatively affect access to and outcomes from transplantation and dialysis outcomes, there is no nationally agreed on approach to screening for these conditions using objective measures during transplant workup or preparation for dialysis in the UK.

Comprehensive Geriatric Assessment (CGA) and optimisation is a multidimensional, multidisciplinary clinical process that addresses the medical, social and psychological needs of older patients to develop an individualised management plan with patients and their carers. It has been in use across different healthcare settings for over 30 years. CGA reduces polypharmacy, hospital admissions, nosocomial infection rates and facilitates timely conversations around advance care planning [15]. In the perioperative setting, CGA and optimisation have demonstrated clinical and cost-effectiveness in both elective and emergency surgery, with reduced postoperative complications and tailored decision-making [15,16,17,18]. The potential to identify and optimise frailty, multimorbidity and cognitive impairment using a transplant-specific CGA and optimisation model prior to transplantation could inform patient selection and improve transplant outcomes in this cohort. A bespoke CGA model for older kidney patients being considered for dialysis, or supportive care, may also be of benefit by mitigating the progression of frailty, optimising the multimorbid burden, identifying/addressing CI and planning ahead through the process of shared decision-making (SDM).

Preliminary work is under way to employ CGA methodology in the care of older CKD patients and potential kidney transplant recipients in several prospective studies across the world [19,20,21,22].

This national UK survey aims to describe:attitudes/beliefs of renal physicians and kidney transplant surgeons in the UK toward the CGA for older potential kidney transplant recipients and older patients who are being considered for dialysis or supportive care;provision of CGA services for these patients in the UK;barriers and enablers to the provision of these CGA services in the UK.

## 2. Materials and Methods

A pair of surveys (28-question survey directed at RU physicians and 21-question survey designed for TC surgeons, included in Appendix A) was developed using themes from the Centre for Perioperative Care—British Geriatrics Society (CPOC-BGS) guidance on perioperative care for people living with frailty undergoing elective and emergency surgery [23] and refined by a working group of nephrologists, transplant surgeons and geriatric medicine consultants.

The working group comprised a convenience sample of relevant stakeholders—two consultant nephrologists from the study centre, one consultant in geriatric medicine and perioperative medicine for older people undergoing surgery (POPS) and two transplant surgeons—from two different transplant centres. The survey was piloted among all nephrologists and transplant surgeons at the study centre, by one nephrologist at a different renal unit (not a transplant centre) and a geriatrician at a different transplant centre. The members of the group outside the study centre were recruited on the basis of their involvement in a previous national working group on frailty in kidney disease.

The survey designed for renal physicians examined practices around frailty, cognitive assessment and discussion of prognosis in more granular detail (Appendix A) and explored CGA for patients over 60 managed with dialysis or receiving supportive care in addition to potential transplant recipients, who were the only group referenced in the survey directed at transplant surgeons.

Multiple-choice, 4-point Likert scale, dichotomous and open-ended questions were included in the survey. Ten expert raters recruited from nephrology, transplant surgery and geriatric and general medicine reviewed the survey for readability and to ensure non-ambiguity. Lawshe’s method was used to calculate the content validity; at 0.88, this value was above the validated threshold of 0.62 for 10 expert raters [24]. A sample of six transplant surgeons and six nephrologists piloted the validated survey; after which, minor edits were made before repiloting.

All 72 renal units in the UK were identified using the UK Kidney Association (UKKA) website. Twenty-three UK centres performing adult kidney transplants were identified using NHS Blood and Transplant (NHSBT) directory data. Surveys were distributed online through the SurveyMonkey platform, using email invitation, directly to the lead consultant nephrologists and transplant surgeons at each centre. The leads were invited to nominate an alternative consultant respondent from their unit if they felt their colleague’s responses would be more representative of departmental practice. This direct contact was initiated in line with evidence-based survey methodology to promote a representative response rate from the sampling frames. The initial invitation was sent on 24 April 2024. A single email reminder was sent one week after this. All centres had returned a response by 31 August 2024.

Basic descriptive statistics were used to analyse the survey results. Emerging themes were identified and catalogued. Formal analysis of the free text responses was not carried out, as such qualitative methodology was beyond the scope of this work, but Table 1 enumerates a selection of free text responses from the transplant centres.

## 3. Results

### 3.1. Response Rate

Responses were received from all 72 UK RUs and all 23 adult kidney TCs (100% response rate). Fifty-four of 72 RU respondents answered all 28 survey questions, while 18 out of 23 TC respondents answered all 21 questions in their survey.

### 3.2. Transplant Activity and Workup

Kidney transplants were performed at 23 out of 72 RUs. Twenty-nine respondents at RUs reported performing preliminary (9/29) or full (20/29) pre-transplant assessments of potential recipients at the unit where they work.

### 3.3. Attitudes, Beliefs and Current Practice Regarding CGA Amongst Renal Physicians and Transplant Surgeons

The vast majority of RUs were in favour of using CGA in kidney patients aged over 60 being considered for transplantation (47/55), being managed with dialysis (52/54) and receiving supportive care (51/54) (Figure 1). Seventeen out of nineteen TCs advocated CGA for those being considered for transplantation (Figure 2).

Most RU respondents described a routine assessment of frailty (33/53) and multimorbidity (50/53) in ESKD patients aged over 60 either pre-dialysis or on dialysis. Where frailty was assessed in RUs, the most frequently used validated tool was the Clinical Frailty Scale (CFS), favoured by 78% of respondents (Figure 3). Fifty-eight percent of respondents used the Mini Mental State Examination (MMSE), while thirty-two percent employed the Montreal Cognitive Assessment (MoCA) to assess cognition at the unit where they work (Figure 4). Where ‘Other’ tools were selected, the majority of respondents reported using clinical judgement or ad hoc assessments of frailty/cognition. Almost half of the RU respondent group believed frailty among dialysis patients over 60 was inadequately addressed by the current NHS services (26/54). Thirteen out of nineteen TC respondents reported a routine assessment of frailty in older potential transplant recipients, but thirteen out of eighteen TCs believed the current NHS services failed to adequately address frailty once diagnosed.

Twenty-eight out of fifty-three RUs reported that cognition was ‘rarely’ or ‘never’ assessed in ESKD patients over 60 at the unit where they work. Many RUs expressed the belief that the current NHS services did not adequately address CI in patients over 60 being considered for transplantation (26/53), managed with dialysis (34/53) or receiving supportive care (19/53). Seven out of nineteen TC respondents reported ‘rarely’ assessing cognition in older potential transplant recipients, and nine out of eighteen believed CI was inadequately addressed by the current NHS services.

Discussion of prognosis by RUs was reported to occur ‘usually’ or ‘always’ with potential transplant recipients over 60 in 48/55 units, with patients over 60 undergoing dialysis in 39/54 units and those receiving supportive care in 52/54 units.

Most respondents did not believe that the unit where they work had a robust method of assessing (RUs 28/54, TCs 10/18) and documenting (RUs 35/54, TCs 10/18) the mental capacity for potential kidney transplant recipients over the age of 60. Similarly, the majority of RUs indicated that the mental capacity of older patients being worked up for dialysis lacked dependable methods of assessment (31/54) and documentation (38/54) in the unit where they work.

### 3.4. Current Availability of CGA and Optimisation Services (CGA-OS) in RUs and TCs

Only six units offered CGA-OS to ESKD patients pre-dialysis or on dialysis aged over 60 (including potential transplant recipients). Where CGA-OS were available, they were offered to potential transplant recipients, dialysis patients and those receiving supportive care.

Geriatricians led CGA-OS in four out of six units, with the remaining two services led by nephrology or an advanced nurse practitioner team. Service delivery occurred in a combined nephrology and geriatric medicine clinic for two units, a specialist geriatric medicine clinic for a further two units and a specialist nephrology clinic for the final two units.

Funding for CGA-OS, where available, came from a specialist charity (2/6), trust funding (2/6), a pilot regional program (1/6) or as part of a regional initiative (1/6).

Most RUs reported regular multidisciplinary meetings (MDMs) to facilitate discussion around potential transplant recipients prior to listing (46/55). All MDMs involved nephrology, 98% involved transplant surgery and 95% involved transplant coordinators. Forty-nine percent had an anaesthesiology presence. One unit reported the attendance of a geriatric medicine specialist at MDM. Twenty-five out of forty-five units reported that potential kidney transplant recipients aged over 60 were automatically discussed at MDM at the unit where they work. The identification of frailty prompted discussion at MDM in 27/45 units.

### 3.5. Barriers and Enablers to the Establishment and Ongoing Provision of CGA-OS

Three key barriers to the establishment and ongoing provision of CGA-OS in RUs were identified: lack of funding (45/51), lack of available staff to deliver CGA (44/51) and time constraints (36/51). Thirty-nine out of fifty-one respondents reported insufficient training on geriatric assessment. TCs reported three main barriers to the establishment and ongoing provision of CGA-OS in TCs: lack of training in CGA (16/18), lack of funding (13/18) and lack of clinical evidence (2/18).

The availability of funding was listed as the most important enabler to the CGA and optimisation service provision by 4/6 units. One respondent believed high-quality training on CGA was the most important enabling factor, and another listed the availability of resources (clinic space/equipment) as the key determining enabler.

Table 1 illustrates free text responses to a question on incentives or support that would encourage respondents to incorporate CGA or optimisation services at their centre.

## 4. Discussion

This survey is the first national study to describe attitudes and beliefs of UK renal physicians from every specialist RU and transplant surgeons from every adult TC towards CGA-OS for older kidney patients. It provides insight into the availability of CGA-OS at UK RUs and identifies barriers and enablers to the establishment and ongoing provision of CGA-OS.

This survey demonstrates that UK clinicians support the provision of CGA services for all older kidney patients, including potential transplant recipients, dialysis patients and those receiving supportive care. However, only six units currently offer CGA-OS. Three key barriers to establishing and sustaining CGA services were identified, namely lack of funding, lack of available staff and lack of protected time to deliver CGA and optimisation.

The impact of frailty, multimorbidity and CI on outcomes for all older patients with ESKD, including potential transplant recipients, was recognised by the respondents. However, assessment and management of frailty and CI in this cohort was identified as an area of shortcoming in the current NHS clinical service provision. At present, dedicated CGA-OS for older CKD patients are available in just a minority of RUs. Reponses suggest that frailty and multimorbidity are being identified during patient review, but provision to treat or optimise these is limited. Over half of the respondent units reported that cognition was rarely assessed. Furthermore, survey respondents were not confident that the centre where they worked had a dependable system for assessing and documenting mental capacity in older potential transplant recipients. These findings warrant further investigation.

Despite recent evidence for integrating geriatrician expertise in pre-transplant evaluation and optimisation [25], reported geriatrician involvement in MDTs among our sample was very low, with only one centre accounting for the geriatrician presence at pre-transplant multidisciplinary discussions.

The use of validated clinical tools to assess and grade frailty and CI is an essential step to facilitate the accuracy of diagnosis, prognosis and management planning for these conditions.

Several clinical tools to objectively measure frailty exist and have been validated for use in older kidney patients. The five-point fried frailty phenotype (FFP), developed in 2001, has been used extensively in kidney patients but lacks a graded measure of frailty and can be challenging to implement in a busy clinical setting. The Short Physical Performance Battery (SPPB) has been used in this population but is based solely on a physical examination and neglects the non-physical elements of frailty syndromes. The CFS based on the deficit accumulation model of frailty is easily employed and, although subjective, represents a global screening tool giving a score that, like that of the FFP, is associated with 12-month mortality in kidney patients [26]. The Edmonton Frail Scale (EFS) assesses nine domains to provide a score out of 17 points, with higher scores indicating greater frailty. The added utility of EFS comes from the inclusion of psychosocial components and documented association between higher scores and poorer patient experience and quality of life among older CKD Stage 5 patients [27].

The MoCA has been validated as a sensitive tool for the early detection of mild CI (MCI) in hundreds of studies across different patient populations since 2000. It is easily implemented in most patients and can identify MCI better than its chief screening competitor, the MMSE, as it addresses visuospatial reasoning and executive function (a deficit that often represents the vascular dementia processes more commonly among ESKD patients) [28]. It may be adjusted for patients with poor literacy or low educational level and has been validated in several languages. Moderate sensitivity (59–80%, depending on degree of impairment) and specificity (50–65%) are reported in the ability of both tools to identify cognitive impairment in CKD patients [29,30], though the MoCA has the better predictive ability for diagnosing severe cognitive impairment [30].

CGA has an evidence-based proven track record in outcome improvement and cost-effectiveness across multiple medical and surgical populations affected by frailty, multimorbidity and CI [15,16,17,18]. A recent systematic review of evidence for partial nephrectomy among older patients with renal malignancy reinforces the importance and benefit of shared decision-making, as well as bespoke holistic optimisation and treatment plans for these individuals [31]. A CGA model adapted to the needs of older ESKD patients could transform outcomes for this cohort and has been advocated by a consortium of European nephrogeriatric experts in a recent narrative review [32].

Though ongoing appraisal of CGA in the setting of ESKD patients over 60 is valid, it is equally important to use the existing evidence base that has demonstrated the clinical and cost-effectiveness of CGA in medical patients, emergency and elective surgical patients and those with cancer to begin to adapt services to the needs of older potential kidney transplant recipients and older patients receiving dialysis or supportive care.

The high prevalence of CI in CKD patients over 60 raises concerns about the decision-making capacity around dialysis and transplant surgery and merits particular attention. Many units lack reliable systems for assessing and documenting capacity, potentially leaving clinicians unsupported and leading to adverse outcomes. For example, without robust frameworks, patients may be deemed unsuitable for transplant or dialysis based solely on impaired decision-making (IDC) rather than the context of IDC’s impact on relevant medical factors. Conversely, failing to assess the capacity could result in inappropriate transplantation, impacting both the patient and other candidates on the waiting list.

Recent publications reviewing practices around SDM in older patients with CI and ESKD have recognised the complexity and heterogeneity of these conversations. There is little agreement on protocolisation or a standardised approach [33,34]. The mental capacity assessment is embedded in the framework of CGA, prompted by the detection of CI during holistic multidomain assessment. Where impaired capacity is identified during preoperative CGA, underlying mechanisms and possible adverse outcomes are explored and mitigated with appropriate influence on shared decision-making [35].

Funding was identified as the most significant barrier to establishing CGA-OS, with lack of staff trained in CGA and time constraints emerging as key challenges. These findings align with the existing literature on the development and implementation of CGA-OS [36]. A recent qualitative study conducted across six NHS hospitals with established Perioperative Medicine for Older People undergoing Surgery (POPS) services in the UK explored strategies employed by clinical leaders to implement POPS services in diverse contexts. The study highlighted barriers such as limited management and financial support, as well as resistance among colleagues across disciplines to adapt to the changes CGA and optimisation bring to traditional workflows [37].

Pilot feasibility studies, stakeholder engagement and development of a compelling business case—demonstrating clinical, operational and financial benefits (e.g., reduced length of stay and lower readmission rates)—were critical in securing support for successfully established programs [37]. Additionally, given the complexity of developing and evaluating interventions like CGA-based services, adopting implementation science approaches is essential for embedding and assessing these services effectively [38].

Geriatrician oversight is crucial. However, considering workforce challenges in geriatric medicine, the long-term vision for CGA in CKD patients emphasises interdisciplinary adaptation [39]. The 2021 UK CPOC-BGS guideline on managing frail patients undergoing emergency and elective surgery offers strategies for developing CGA-based perioperative services and proposes metrics for evaluation [22]. By fostering collaboration across specialties, CPOC aims to improve perioperative care, making the representation of transplant-specific issues within such an organisation essential. This focus is vital to addressing the unique challenges faced by older potential kidney transplant recipients and leveraging the opportunities CGA presents, as highlighted in this survey.

Workforce and training reforms are critical for the successful implementation and sustainability of CGA-OS [38]. Dedicated education on these topics, where possible integrating clinical exposure, should be incorporated into specialty training curricula of all professionals caring for older patients with CKD.

In the UK, the BGS has developed a comprehensive curriculum for POPS, structured around four adaptable domains: Knowledge, Skills, Behaviours and Specific Learning Methods [39]. This framework offers a valuable model for embedding CGA and optimisation training.

Efforts are underway in the UK to develop and integrate a CKD patient-specific CGA tool into clinical practice. The goal is to enhance patient selection and improve post-transplant outcomes for potential transplant recipients over 60. This initiative builds on the expanding body of evidence supporting the use of CGA for older surgical patients in various specialties across the US [40], UK [16] and globally [17].

The inherent limitations of survey research are present, including an incomplete response or response bias, which was mitigated by achieving a 100% response rate, and ambiguity, which was reduced through piloting with renal physicians and transplant surgeons in addition to validation by a group of expert raters. Direct contact with clinical leads with the option to nominate an alternative colleague to complete the survey on behalf of their unit ensured representative responses from the sampling frame within the limits of survey methodological best practice.

## 5. Conclusions

This survey, the first of its kind in the UK, demonstrates strong stakeholder support for CGA-OS targeting older ESKD patients, including potential transplant recipients, dialysis patients and those receiving supportive care. Currently, very few UK RUs offer CGA-OS for these patient cohorts. The survey identifies three key barriers to developing and implementing these services: lack of funding, insufficient trained staff and time constraints. However, we believe these challenges can be overcome, as demonstrated in other medical and surgical settings, where CGA has proven clinically and cost-effective.

Further research demonstrating the benefits of CGA-OS for older ESKD patients is imperative. This survey represents a significant step toward feasibility studies on adapted CGA for the older kidney patient.

## Figures and Tables

**Figure 1 jcm-14-03070-f001:**
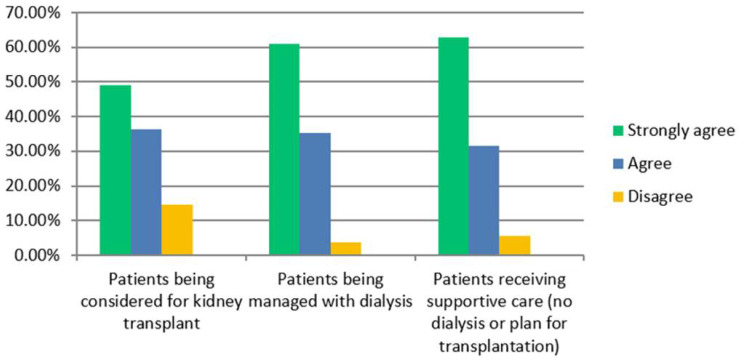
Renal units. Please indicate the extent to which you agree that there is a role for CGA in older kidney patients in each of the listed scenarios.

**Figure 2 jcm-14-03070-f002:**
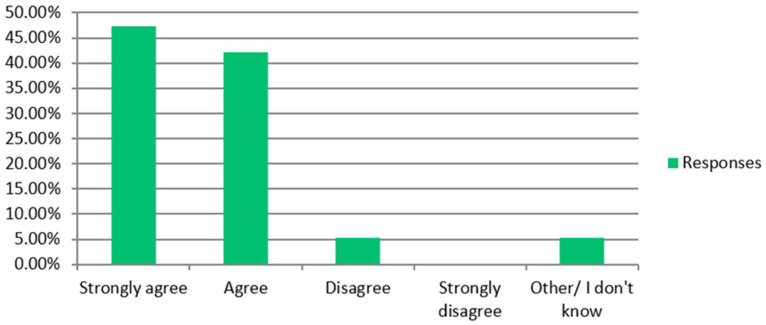
Transplant centres. Please state the extent to which you agree that there is a role for CGA in patients over 60 being considered for transplantation.

**Figure 3 jcm-14-03070-f003:**
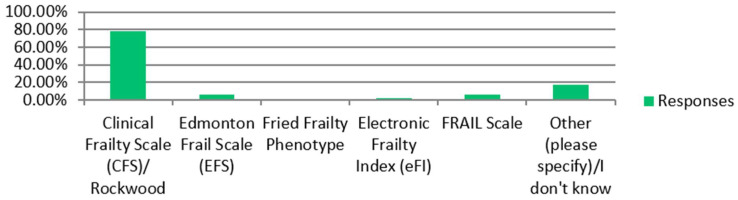
Renal units. Choice of validated clinical tool used to measure frailty.

**Figure 4 jcm-14-03070-f004:**
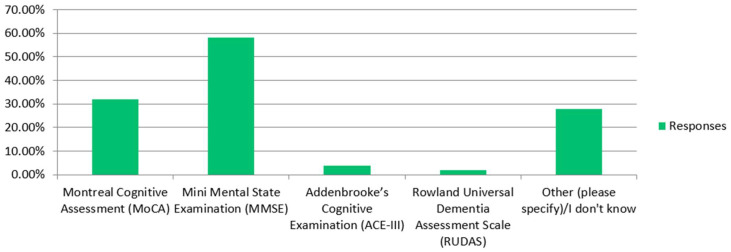
Renal units. Choice of validated clinical tool to measure cognition.

**Table 1 jcm-14-03070-t001:** Transplant centres. Free text responses.

What Incentives or Support Would Encourage You to Incorporate Comprehensive Geriatric Assessment and Optimisation (CGA) More Consistently in the Evaluation of Potential Kidney Transplant Recipients Aged over 60?
‘Really strong evidence that it improves outcomes in this population.’
‘Finance and work force.’
‘Incorporate it into recipient assessment.’
‘Improved evidence on frailty specifically in kidney transplantation.’
‘Evidence-base.’
‘An interested Care of the Elderly Physician and the resources to collaborate with them- time, financial and logistics.’
‘Education on the process and funding. Unfortunately, unlikely to be able to set anything new up which will cost money due to the financial stress the hospital is under.’
‘Education from centres that have used CGA, evidence of better outcomes when using CGA.’
‘Evidence, funding.’

## Data Availability

The raw data supporting the conclusions of this article will be made available by the authors on request.

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
