# Peer review of "Comprehensive Geriatric Assessment (CGA) and Optimisation Services in Older Kidney Patients: Results from the First UK-Wide Transplant Centre and Renal Unit Survey Study"

_jcm, 2025, doi:10.3390/jcm14093070_

Round 1

Reviewer 1 Report

Comments and Suggestions for Authors

The survey-based study encompasses all 72 renal units and 23 transplant centres in the UK, which is an exceptional and rare 100% response rate (Wow!)

  • Clarify why only 6 units have CGA-OS despite such high clinician support—was it solely infrastructure or also cultural/knowledge barriers?
  • Discuss variability in cognitive assessment tools (MMSE vs MoCA) in more depth—are some being misused or underutilised?
  • Can you prorpose a model implementation framework (even schematic) for renal CGA-OS services?
  • Con you report examples or outcomes from the 6 centres where CGA is already operational?
  • Please comment on the lack of geriatrician involvement in transplant MDTs despite clinical recommendations.

To elevate the translational potential of your work, I strongly encourage you to strengthen the discussion by:

  • Proposing a practical framework for CGA-OS integration, possibly inspired by existing perioperative geriatrics models.
  • Highlighting clinical outcomes or early lessons from the six UK units already delivering these services.
  • Emphasising the structural absence of geriatricians in transplant MDTs — a disconnect that risks perpetuating suboptimal assessment and shared decision-making in a high-risk cohort.

In this context, it would be highly appropriate to reference recent surgical-oncologic literature which reinforces the same message: age is not the barrier — frailty is. For example, the systematic review - EJSO, 2024; PMID: 39121634 -  explores outcomes of partial nephrectomy in elderly patients with renal cancer and demonstrates how oncologic decision-making is being reshaped by geriatric assessment principles, not chronological age alone. Their call for structured frailty evaluation before oncological surgery mirrors your findings and further justifies the urgency of CGA-OS implementation in the renal-transplant space.

Comments on the Quality of English Language

good

Author Response

To elevate the translational potential of your work, I strongly encourage you to strengthen the discussion by:

  1. Proposing a practical framework for CGA-OS integration, possibly inspired by existing perioperative geriatrics models.
  2. Highlighting clinical outcomes or early lessons from the six UK units already delivering these services.
  3. Emphasising the structural absence of geriatricians in transplant MDTs — a disconnect that risks perpetuating suboptimal assessment and shared decision-making in a high-risk cohort.
  4. In this context, it would be highly appropriate to reference recent surgical-oncologic literature which reinforces the same message: age is not the barrier — frailty is. For example, the systematic review - EJSO, 2024; PMID: 39121634 - explores outcomes of partial nephrectomy in elderly patients with renal cancer and demonstrates how oncologic decision-making is being reshaped by geriatric assessment principles, not chronological age alone. Their call for structured frailty evaluation before oncological surgery mirrors your findings and further justifies the urgency of CGA-OS implementation in the renal-transplant space.

Response:

Thank you for these suggestions. Each is addressed in the responses below. 

We will also reference the paper by Lasorsa et al from EJSO (10.1016/j.ejso.2024.108578). Thank you for mentioning this- it is an excellent example of CGA principles being applied to a complex older oncology cohort with good adapted treatment approaches and critically important pre-op identification of frailty.

Comment 1: 

The survey-based study encompasses all 72 renal units and 23 transplant centres in the UK, which is an exceptional and rare 100% response rate (Wow!)

Clarify why only 6 units have CGA-OS despite such high clinician support—was it solely infrastructure or also cultural/knowledge barriers?

Response:

We thank the reviewer for taking the time to read our paper and for their detailed comments- these insights are greatly appreciated.

The mismatch between the enthusiasm expressed by most respondents for CGA-OS and the limited availability of CGA is certainly a point worth addressing- while our survey was not designed to interrogate individual unit reasons for not having an active CGA-OS initiative and we would not wish to speculate, we do describe what respondents consider to be the barriers to CGA setup and implementation. In this work these were identified as lack of funding, lack of available staff, and lack of protected time to deliver CGA and optimisation.

Comment 2: 

Discuss variability in cognitive assessment tools (MMSE vs MoCA) in more depth—are some being misused or underutilised?

Response:

Cognitive assessment forms an integral element of CGA, and validated tools such as MoCA or MMSE are helpful in performing these assessments. We compare their utility in the kidney patient cohort within our discussion. The purpose of this study was to assess attitudes and beliefs among UK nephrologists and transplant surgeons towards CGA-OS for older kidney patients, and to describe current CGA availability in all UK renal units/transplant centres, and whilst we would like additional detail on the appropriate use of relevant cognitive assessment tools this was beyond the scope of this study. These surveys set the scene for upcoming work on feasibility-testing a kidney transplant-specific CGA-OS alluded to in the conclusion which will also consider which tools should be used as you suggest.

Comment 3:

Can you propose a model implementation framework (even schematic) for renal CGA-OS services?

Response:

Thank you for this suggestion. We agree that this will be key in helping units establish services as the literature base in support of CGA-OS grows. Unfortunately, detail on this is beyond the scope and word count of this survey research but we agree with the reviewer and have cited relevant MRC guidance on developing and evaluating complex interventions to acknowledge this point.

Comment 4:

Con you report examples or outcomes from the 6 centres where CGA is already operational?

Response:

Regrettably, we do not have outcome data from the centres where CGA-OS are up and running but we have referenced a recent narrative review on CGA-OS in older kidney patients (https://doi.org/10.3390/jcm14051749) and other literature exploring CGA-OS outcomes in other groups.  We hope that studies such as these and the work we are doing to facilitate collaboration between units will lead to outcomes data being published across centres.

Comment 5:

Please comment on the lack of geriatrician involvement in transplant MDTs despite clinical recommendations.

 Response:

Thank you for this comment. We aren’t aware of the guidelines which refer to geriatrician involvement in transplant MDMs. It may be that the expert reviewer is referring to less specific surgical guidelines which for several years now have advocated preoperative CGA and involvement of geriatric medicine expertise throughout the pathway of care for older surgical patients (ACS NQIP/AGS 2012, Access all ages 2011, NCEPOD age old problem 2010, CPOC/BGS 2021 etc) but to our knowledge KDIGO and UKKA don’t have formal statements on geriatrician presence at transplant listing MDTS.

We have acknowledged your suggestion through referencing this paper from Transplant International in 2023- https://doi.org/10.3389/ti.2023.11296 where we discuss the lack of geriatrician involvement in MDTs. It appears in the manuscript as below:

Despite recent evidence for integrating geriatrician expertise in pretransplant evaluation and optimisation (https://doi.org/10.3389/ti.2023.11296), reported geriatrician involvement in MDTs among our sample was very low, with only one centre accounting for geriatrician presence at pre-transplant multidisciplinary discussions.

Reviewer 2 Report

Comments and Suggestions for Authors

This first UK national study showed that currently very few UK RUs offer CGA-OS and therefore lacked reliable systems for assessing and documenting capacity thus potentially risk adverse outcomes. It also proves that UK clinicians support the provision of CGA services for all older kidney patients. Study identifies key barriers and enablers to the establishment and ongoing provision of CGA-OS. Authors also provide limitations of the survey.  It reminds us that comprehensive care of the patient must encompass patient oriented outcomes.

Author Response

Comments:

This first UK national study showed that currently very few UK RUs offer CGA-OS and therefore lacked reliable systems for assessing and documenting capacity thus potentially risk adverse outcomes. It also proves that UK clinicians support the provision of CGA services for all older kidney patients. Study identifies key barriers and enablers to the establishment and ongoing provision of CGA-OS. Authors also provide limitations of the survey.  It reminds us that comprehensive care of the patient must encompass patient oriented outcomes.

Response:

We grateful acknowledge the comments made by the reviewer. Thank you for taking the time to read out paper and for your insights.

Reviewer 3 Report

Comments and Suggestions for Authors

Dear authors,

  1. This paper achieves a 100% response rate, but it is limited to only 72 renal units (RUs) and 23 transplant centers (TCs). Since the survey focuses on these units and centers, the results may not be nationally representative. In particular, there may be a lack of data from rural or smaller hospitals, so expanding the sample to include a broader and more diverse range of settings would increase the reliability of the findings. This should be reconsidered.

  1. The paper primarily presents quantitative survey results, but it lacks detailed qualitative data regarding the specific barriers and enablers to CGA-OS provision. Additional interviews or case studies could provide deeper insight into the real-world challenges and success stories of each unit, complementing theoretical understanding with practical perspectives.

  1. While the paper mentions the use of the Mini Mental State Examination (MMSE) and Montreal Cognitive Assessment (MoCA) for assessing cognitive function, it does not sufficiently discuss which tools are most effective. A more detailed comparative study of cognitive impairment (CI) assessment tools, along with an evaluation of their applicability to specific patient populations, is needed.

  1. The paper highlights that CGA-OS services are offered in only a few units, but it does not provide an in-depth strategic analysis of the underlying factors contributing to success or obstacles. It would be helpful to present specific methodologies for securing funding and staffing, which are critical for expanding services. Additionally, more attention should be given to the strategies for maintaining services, such as funding mechanisms and collaboration with governments and organizations.

  1. While the paper mentions a lack of training as a barrier, education for healthcare professionals is key to promoting the implementation of CGA. A more detailed description of training programs, including content and delivery methods, would clarify the specific steps needed for effective implementation.

  1. The research focuses on a small number of units, but a nationwide, multi-center comparative study would help identify regions and institutions where CGA-OS is not yet available, as well as operational challenges across different healthcare settings.

  1. The paper makes policy recommendations, but it lacks a concrete strategy for how these recommendations would influence actual policy changes. For example, it would be helpful to provide a roadmap for promoting the adoption of CGA-OS, such as lobbying the government, introducing funding assistance programs, or other detailed policy initiatives to support the implementation of these services.

Author Response

Comment 1:

Dear authors,

 This paper achieves a 100% response rate, but it is limited to only 72 renal units (RUs) and 23 transplant centers (TCs). Since the survey focuses on these units and centers, the results may not be nationally representative. In particular, there may be a lack of data from rural or smaller hospitals, so expanding the sample to include a broader and more diverse range of settings would increase the reliability of the findings. This should be reconsidered.

Response:

We gratefully acknowledge the comments made by the reviewer and thank them for taking the time to read our paper. These remarks are invaluable in helping us to refine our work.

Thank you for this comment. Each of the UK’s 72 hospitals with a formal nephrology department and affiliated dialysis units, in addition to all 23 UK centres where adult kidney transplants are performed, were represented in this study. This, we feel, yields representative data from which findings can be reasonably inferred as inviting all practicing UK nephrologists and transplant surgeons would not be feasible.

Comment 2:

The paper primarily presents quantitative survey results, but it lacks detailed qualitative data regarding the specific barriers and enablers to CGA-OS provision. Additional interviews or case studies could provide deeper insight into the real-world challenges and success stories of each unit, complementing theoretical understanding with practical perspectives.

Response:

This is an excellent point, thank you. We invited units to contact us if they were interested in pursuing a national initiative on CGA-OS for older kidney patients, and will follow up with those who have reached out. This will allow us to publish further on individual centre experiences with CGA.

Comment 3:

While the paper mentions the use of the Mini Mental State Examination (MMSE) and Montreal Cognitive Assessment (MoCA) for assessing cognitive function, it does not sufficiently discuss which tools are most effective. A more detailed comparative study of cognitive impairment (CI) assessment tools, along with an evaluation of their applicability to specific patient populations, is needed.

Response:

We agree with this point. Data on optimal cognitive assessment tools for kidney patients is limited. We comment on performance of MoCA vs MMSE in distinct CKD populations in our discussion, but robust data on comparative performances of available cognitive testing tools is needed. Of course, making a diagnosis of cognitive impairment requires a threshold score on a validated assessment tool in addition to clinical identification of a functional deficit per DSM-V criteria.

Comment 4:

The paper highlights that CGA-OS services are offered in only a few units, but it does not provide an in-depth strategic analysis of the underlying factors contributing to success or obstacles. It would be helpful to present specific methodologies for securing funding and staffing, which are critical for expanding services. Additionally, more attention should be given to the strategies for maintaining services, such as funding mechanisms and collaboration with governments and organizations.

Response:

Thank you- this is another point with which we agree. While in-depth discussion of underlying factors determining the success and sustainability of CGA-OS (including funding and staffing) is beyond the scope of this paper, future implementation/organisational science research should focus on logistical components of establishing and sustaining CGA-OS for older kidney patients. This research will need to draw on existing data in perioperative medicine and oncology given the dearth of studies on CGA-OS for older kidney patients.

This issue is reflected in the manuscript:

A recent qualitative study conducted across six NHS hospitals with established Peri-operative Medicine for Older People undergoing Surgery (POPS) services in the UK explored strategies employed by clinical leaders to implement POPS services in diverse contexts. The study highlighted barriers such as limited management and financial support, as well as resistance among colleagues across disciplines to adapt to the changes CGA and optimisation bring to traditional workflows. [38]

Pilot feasibility studies, stakeholder engagement, and development of a compelling business case—demonstrating clinical, operational, and financial benefits (e.g. reduced length of stay, lower readmission rates)—were critical in securing support for successfully established programs. [39,40] Additionally, given the complexity of developing and evaluating interventions like CGA-based services, adopting implementation science approaches is essential for embedding and assessing these services effectively. [41]

Comment 5:

While the paper mentions a lack of training as a barrier, education for healthcare professionals is key to promoting the implementation of CGA. A more detailed description of training programs, including content and delivery methods, would clarify the specific steps needed for effective implementation.

Response:

Detailed description of CGA-OS educational modules and training plans aimed at junior clinical stakeholders is a very important element of an implementation plan for CGA-OS. The purview of this current paper precludes a step-by-step breakdown of such programs/material but will be the focus of dedicated literature in the future.

Comment 6:

The research focuses on a small number of units, but a nationwide, multi-center comparative study would help identify regions and institutions where CGA-OS is not yet available, as well as operational challenges across different healthcare settings.

Response:

Thank you, we agree that it is important to represent the interests and resource capabilities of all UK healthcare settings in research on emerging clinical initiatives.

Comment 7:

The paper makes policy recommendations, but it lacks a concrete strategy for how these recommendations would influence actual policy changes. For example, it would be helpful to provide a roadmap for promoting the adoption of CGA-OS, such as lobbying the government, introducing funding assistance programs, or other detailed policy initiatives to support the implementation of these services.

Response:

We thank the reviewer for this thoughtful and important suggestion. We agree that a detailed roadmap to inform national policy would be a valuable contribution to this field.

There is a balance required between demonstrating that CGA has clinical and cost effectiveness in those with CKD 5 being considered for dialysis and transplant listing and avoiding the situation where an intervention (CGA and optimisation) which has demonstrated benefit in numerous other frail multimorbid groups of older patients (oncology/perioperative/ admitted with acute medical issues) is withheld whilst studies are repeated to show the same benefit.

Future work will be needed to inform such a roadmap, including pilot feasibility studies of CGA-OS delivery and outcome evaluation in line with the MRC framework for developing and evaluating complex interventions (Skivington et al., 2021) but this should be done alongside implementation science work to describe and address barriers to implementation avoiding a protracted implementation gap as is often observed. Once feasibility and benefit are demonstrated, integration into national renal service specifications, commissioning guidance, GIRFT recommendations, and NHS benchmarking frameworks should be explored.

Our manuscript reflects this discussion in the following paragraph:

Efforts are underway in the UK to develop and integrate a CKD patient-specific CGA tool into clinical practice. The goal is to enhance patient selection and improve post-transplant outcomes for potential transplant recipients over 60. This initiative builds on the expanding body of evidence supporting the use of CGA for older surgical patients in various specialties across the US [44], UK [16], and globally. [17]

Round 2

Reviewer 3 Report

Comments and Suggestions for Authors

Dear Authors,

I am satisfied with the revisions that have been made by the authors.